# The Dual Role of Circular RNAs as miRNA Sponges in Breast Cancer and Colon Cancer

**DOI:** 10.3390/biomedicines9111590

**Published:** 2021-10-31

**Authors:** Jiashu Huang, Shenghao Yu, Lei Ding, Lingyuan Ma, Hongjian Chen, Hui Zhou, Yayan Zou, Min Yu, Jie Lin, Qinghua Cui

**Affiliations:** 1Lab of Biochemistry & Molecular Biology, School of Life Sciences, Yunnan University, Kunming 650091, China; huangjiashu_666@163.com (J.H.); ysh1042939168@163.com (S.Y.); dingleiynu@ynu.edu.cn (L.D.); malingyuan15@163.com (L.M.); chenhj098@163.com (H.C.); zhouhui201306@163.com (H.Z.); zouyayan972@163.com (Y.Z.); yumin@ynu.edu.cn (M.Y.); linjie@ynu.edu.cn (J.L.); 2Key Lab of Molecular Cancer Biology, Yunnan Education Department, Kunming 650091, China

**Keywords:** circRNA, miRNA sponge, breast cancer, colorectal cancer

## Abstract

Breast cancer (BC) and colon cancer (CRC) are the two most deadly cancers in the world. These cancers partly share the same genetic background and are partially regulated by the same genes. The outcomes of traditional chemoradiotherapy and surgery remain suboptimal, with high postoperative recurrence and a low survival rate. It is, therefore, urgent to innovate and improve the existing treatment measures. Many studies primarily reported that the microRNA (miRNA) sponge functions of circular RNA (circRNA) in BC and CRC have an indirect relationship between the circRNA–miRNA axis and malignant behaviors. With a covalent ring structure, circRNAs can regulate the expression of target genes in multiple ways, especially by acting as miRNA sponges. Therefore, this review mainly focuses on the roles of circRNAs as miRNA sponges in BC and CRC based on studies over the last three years, thus providing a theoretical reference for finding new therapeutic targets in the future.

## 1. Introduction

According to statistics from the International Agency for Research on Cancer (IARC) of the World Health Organization (WHO), breast cancer (BC) is the second most common type of tumor, with about 18.1 million new cases reported worldwide in 2018. In women, BC has the highest prevalence in almost all countries and regions [1]. BC is highly heterogeneous and depends on the activity of human epidermal growth factor receptor 2 (HER2), the estrogen receptor (ER), and the progesterone receptor (PR) [2]. Triple-negative breast cancer (TNBC, ER, PR, and HER2 are negatively expressed) contributes to the high mortality of BC patients since it is closely associated with the high level of metastasis [3,4]. The lack of targeted medicine leaves chemotherapy as the main treatment for TNBC [5,6]. However, such treatment is unsatisfactory and often causes undesirable side effects [7,8].

Colon cancer (CRC) is the second leading cause of death after lung cancer in the United States [9], with a risk ranking of fourth in the world for people under the age of 75 [10]. In 2020, approximately 147,950 individuals were diagnosed with CRC and 53,200 died from the disease in the United States [11]. Recent studies found that people under the age of 50 suffering from CRC are more likely to suffer from aggressive advanced tumors [12]. Clinical data indicate that the broad-spectrum chemotherapy drugs used to treat CRC include 5-fluorouracil (5-FU), oxaliplatin, and irinotecan and that the median survival of the patients with these drugs is 18 to 20 months. Vascular endothelial growth factor (VEGF) inhibitors and epidermal growth factor receptor (EGFR) inhibitors are often used in combination with the above drugs to increase the median survival of the patients to 30 months [13]. However, adjuvant and multi-dose adjuvant chemotherapy have no effects on patients with stage II, III, or IV cancer and may even damage patients’ retained rectal functions [14,15,16]. Studies showed that 10–20% of CRC patients had a positive family history of the disease [17,18], with the heritability among CRC patients of about 35% [19]. In malignant tumors, four or five mutation genes are required for benign tumor transformation [20]. Many oncogenes are involved in tumorigenesis and development, including the *KRAS* proto-oncogene (*KRAS* is a Kirsten *RAS* oncogene homolog from the mammalian *RAS* gene family that encodes proteins implicated in various malignancies), *EGFR*, the *GNAS* complex locus (*GNAS* enables adenylate cyclase activator activity and regulates cell growth and cell division), the kinase insert domain receptor (*KDR*), the B-Raf proto-oncogene (*BRAF*, serine/threonine kinase), the phosphatidylinositol-4,5-bisphosphate 3-kinase catalytic subunit alpha (*PIK3CA*), and resistance to audiogenic seizures (*R**AS*), as well as tumor-suppressor genes, including adenomatous polyposis coli (*APC* encodes a tumor-suppressor protein that acts as an antagonist of the Wnt signaling pathway), tumor protein p53 (*TP53* encodes tumor protein p53), N-myc downstream-regulated 1 *(NDRG1)*, the putative oncoprotein nm23 *(NM23)*, cadherin 1 *(CDH1)*, Wnt/β-catenin, transforming growth factor beta (*TGF-β*)/SMAD family member (*SMAD*), nuclear factor kappa B (NF-κB), and the Notch and Janus kinases (*JAKs*)/ signal transducer and activator of transcription 3 (*STAT3*) signaling pathways [21,22,23].

Some studies showed that BC and CRC partly share the same genetic backgrounds and are also partly regulated by the same tumor-suppressor genes during tumor initiation, proliferation, and metastasis [21,24,25,26]. Utilizing databases from the Cancer Genome Atlas (TCGA) and National Center for Biotechnology Information (NCBI), we found that some oncogenes regulate the proliferation and metastasis in both BC and CRC, including *BRAF*, *PIK3CA*, erb-b2 receptor tyrosine kinase 2 *(ERBB2)*, *TP53*, the phosphatase and tensin homolog *(PTEN)*, fibroblast growth factor receptor 1 *(FGFR1)*, erb-b2 receptor tyrosine kinase 3 *(ERBB3)*, the kinase insert domain receptor (*KDR*) and proteasome 26S subunit ubiquitin receptor, and non-ATPase 4 *(PSMD4)*, suggesting the existence of common regulatory mechanisms between these two cancers. Therefore, using gene-targeted therapy to regulate the proliferation, metastasis, and invasion of BC and CRC tumors may provide new hope in the treatment of these two cancers in the future and have great significance in increasing the survival rate and improving the prognosis of patients.

## 2. Biogenesis and Functions of circRNA

Unlike traditional linear RNA (containing 5′ and 3′ ends), circular RNA (circRNA) is a type of non-coding RNA that does not own a 5′ end cap or a 3′ end poly(A) tail whose circular structure is formed by covalent bonds [27]. Structurally, the circRNA molecule exhibits a closed loop that is not affected by RNA exonuclease (RNA Exo) and is stably expressed in the body [28]. This molecule was first discovered in 1976 during a study on potato spindle tuber disease, wherein the researchers found that viroids can infect plants and cause death [29]. Unlike a virus, a viroid has no protein coating, and its genome is a single-stranded closed RNA molecule [29,30]. Afterwards, circRNA was found in eukaryotic cells through electron microscopy [31]. Subsequently, thousands of well-expressed and stable circRNAs were detected during the sequencing and computational analysis of RNA from humans, mice, and nematodes [32].

According to biogenesis, the types of circRNA mainly include (see Figure 1 for details) (a) exonic circRNA, the most common type of circRNA, connects the 5′ donor site downstream of the exon to the upstream 3′ acceptor site to form a cable-tail plug and circularization and then cuts to remove introns, resulting in circRNA [33,34]; (b) circRNA combined with introns and exons, also known as elcircRNA, for which the reverse complement (cis-acting element) is located in the intron flanks and promotes circRNA biogenesis by pairing to form RNA duplexes that juxtapose the splice sites followed by alternative splicing [35]; (c) circRNA formation is regulated by RNA-binding proteins (RBPs), such as the KH domain containing RNA binding (QKI), FUS RNA-binding protein (FUS), and RNA-binding motif protein 20 (RBM20) [36]; (d) lariat-type circRNA is composed of introns, spliceosome-dependent back-splicing of introns, and the splicing of introns [33].

Recent studies have shown that the regulation of miRNA on target genes is largely controlled by circRNA [37]. Large numbers of miRNA binding sites on circRNA compete with messenger RNA (mRNA), which is known as competitive endogenous RNA (ceRNA) and habitually called a miRNA sponge [38]. For example, the circular RNA sponge of miR-7 (CIRs-7), which is highly expressed in mouse and human brains, structurally contains more than 70 miRNA-conserved binding sites, enabling it to significantly inhibit the activity of miR-7 and upregulate the expression levels of target genes [39]. Due to circRNA’s specific high-level inhibitory effects on miRNAs, there has been significant progress in using circRNA to regulate disease-related miRNAs and downstream target genes. For instance, circ_100146, which is highly expressed in non-small-cell lung cancer (NSCLC), acts as an miRNA sponge to regulate the expression of multiple downstream mRNAs of splicing factor 3b subunit 3 (*SF3B3*) by binding miR-361-3p and miR-615-5p, thus inhibiting the proliferation and invasion of NSCLC tumors and promoting cell apoptosis [40]. The high expression of circ_100338 in hepatocellular carcinoma (HCC) clearly relates to the low survival rate and high degree of metastasis among HCC patients. In addition, circ_100338 works as an endogenous sponge for miR-141-3p in HCC by causing miR-141-3p to regulate the invasion and migration of HCC [41]. Taken together, circRNA plays a significant role in various types of tumors via the miRNA sponge function. Thus, the main purpose of this review is to explore the role of the miRNA sponges of circRNA on malignant characterizations, such as the growth and metastasis of BC and CRC.

## 3. The Dual Role of circRNA as miRNA Sponges in BC

Multiple studies showed that circRNA was involved in BC based on the results from BC tissues and/or cell lines. It is accepted that the involvement of circRNA in BC is context-specific due to evidence indicating that circRNAs play a dual role in tumor-suppressing and tumor-promoting activities.

### 3.1. Oncogenic circRNA as miRNA Sponges in BC

Oncogenic circRNAs mainly target and inhibit the expression of endogenous tumor-suppressor genes and accelerate carcinogenesis. The upregulation of oncogenic circRNAs promotes progression and is positively correlated with the clinical stage of BC via disturbing normal genes. Primarily, upregulated circRNAs absorb and downregulate the expression of tumor-suppressive miRNAs, leading to the increased expression of the target oncogenic genes that cause proliferation, metastasis, and invasion, as well as accelerating the cell cycle and cell energy metabolism.

In the cell viability, cell cycle, and cell apoptosis in BC and TNBC, circRNAs are involved. Liu et al. reported that circ_GNB1 sponged miR-141-5p and facilitated TNBC cell viability by upregulating the oncogenic gene insulin-like growth factor 1 receptor (*IGF1R*) in a xenograft model [42]. Circ_0000518 was first identified as a sponge of miR-326, which was found to accelerate the cell cycle by targeting fibroblast growth factor receptor-1 (*FGFR1*) in BC [43]. Circ_0008039 is located at chr7:716865-751164, 462 nucleotides in length, and derived from the back-splicing of the protein kinase cAMP-dependent type I regulatory subunit beta (*PRKAR1B*) transcript (NM_001164761). Liu et al. indicated that circ_0008039 could accelerate the transition of the cell cycle from the G0/G1 to the S phase and promote proliferation through the miR-432-5p/E2F transcription factor 3 (*E2F3*) axis in BC [44]. Circ_ABCC4 was found to enhance BC cell activity and inhibited apoptosis by sponging miR-154-5p. In a previous study, circ_ABCC4 also targeted NF-κB and activated the Wnt/β-catenin pathway [45]. Circ_TFF1 suppressed the development of BC by accelerating the apoptosis of BC cells via the miR-338-3p/*FGFR1* axis [46]. In addition, the abnormal expression of several circRNAs in BC, such as circ_0007534 [47], circ_PLK1 [48], and circ_TP63 [49], were shown to be related to cell behaviors and tumorigenesis, such as promoting apoptosis and regulating cell viability and the cell cycle process.

Moreover, circRNAs are involved in the cell migration and invasion in BC and TNBC. Circ_HIPK3, derived from the exon 2 splicing of the homeodomain interacting protein kinase 3 (*HIPK3*), sponges miR-193a and eliminates the inhibition of high mobility group protein box 1 (*HMGB1*) through the phosphatidylinositol 3-kinase (*PI3K*)/AKT serine/threonine kinase (*AKT*) signaling pathway involved in regulating the migration and invasion of BC cells [50]. Xing et al. confirmed that the circ_0005571 (circ_IFI30) generated from head-to-tail splicing functions as a miRNA sponge. The overexpression of circ_IFI30 was found to increase the expression of the CD44 molecule (CD44), which is a transmembrane glycoprotein involved in cell proliferation, differentiation, adhesion, and migration. The decreased expression of twist family bHLH transcription factor 1 (*TWIST1*), zinc finger E-box binding homeobox 1 (*ZEB1*), and E-cad, as well as the downregulation of circ_IFI30, played different roles in TNBC in a xenograft model via the circ_IFI30/miR-520b-3p/CD44 axis in TNBC cells [51]. Circ_BACH2 is derived from the transcription factor BTB domain and CNC homolog 2 (*BACH2*), which is responsible for the differentiation of innate and adaptive cellular lines [52]. The transcriptional program of antibody class switching involves the repressor *BACH2* [53]. The upregulation of circ_BACH2 was associated with the T, N, and tumor node metastasis (TNM) stages. Circ_BACH2 sponges miR-186-5p and miR-548c-3p to regulate the expression of the C-X-C motif chemokine receptor 4 (*CXCR4*), thereby promoting the proliferation and metastasis of TNBC cells [54]. *ZEB1*, as a transcription factor, is widely regarded as an important driving factor behind tumor growth and metastasis. *ZEB1* directly binds to the WWC family member 3 (*WWC3*) promoter to increase the expression of *WWC3* precursor mRNA, followed by reverse splicing to form circ_WWC3. Circ_WWC3 upregulates the expression of multiple oncogenes in the RAS-signaling pathways by absorbing miR-26b-3p and miR-660-3p [55]. Several studies reported that other circRNAs also promoted cell migration and invasion; these circRNAs include circ_0011946 [56], circ_MMP11 [57], circ_0000291 [58], circ_MYO9B [59], and circ_RAD18 [60].

Further, circRNAs are involved in the energy metabolism in BC and TNBC. A growing number of studies show that most cancer cells prefer to use adenosine triphosphate (ATP) through aerobic glycolysis rather than phosphorylation to meet their metabolic needs to maintain cell proliferation, even under aerobic conditions [61,62]. Therefore, the inhibition of aerobic glycolysis is an effective therapeutic strategy to hinder cancer malignancy [63]. Circ_0072088 (circ_ZFR) acts as a carcinogenic regulator in BC cells. Based on a loss of function in vitro and in vivo, Chen et al. first reported an association between low levels of circ_ZFR, glycolysis, and cell proliferation. Mechanistically, circ_ZFR sponges miR-578 to regulate the expression of hypoxia-inducible factor 1 subunit alpha (*HIF1A*), thereby suppressing malignant BC progression via the regulation of glycolysis [64]. Chuang et al. found that silencing circ_0072995 decreased the glucose uptake, lactate production, ATP levels, enzymes of hexokinase 2 (*HK-2*), and lactate dehydrogenase A (*LDHA*), as well as Glucose transporter 1 *(Glut1)*, leading to the suppression of aerobic glycolysis in BC cells through the miR-149-5p/serine hydroxymethyltransferase 2 (*SHMT2*) axis, suggesting that circ_0072995 functions as an oncogenic circRNA and promotes BC progression by inducing malignant cell phenotypes and anaerobic glycolysis [65]. Several studies reported that other circRNAs also play vital roles in regulating energy metabolism, such as circ_0101187 (circYY1) [66] and circ_TFF1 [46].

In addition to promoting malignant biological behavior, circRNA has also been reported to affect stem cell activity, angiogenesis, and drug resistance. Circ_DCAF6 upregulates the GLI family zinc finger 1 (*GLI1*) by sponging miR-616-3p to activate the Hh-signaling pathways and promote stem cell activity [67]. Liu et al. reported that circ_002178 positively regulates collagen type I alpha 1 chain (*COL1A1*) by sponging miR-328-3p, which is conducive to energy metabolism, angiogenesis, and the aggressive malignant behaviors of BC cells [68]. Dou et al. reported that a knockdown of circ_UBE2D2 reduced doxorubicin resistance by facilitating doxorubicin-induced apoptosis through the modulation of miR-512-3p and the cell division cycle associated 3 (*CDCA3*) signal pathway [69].

Taken together, many upregulated circRNAs exert functions as oncogenes through upregulating multiple target genes (Table 1), enabling circRNAs to serve as a potential diagnostic marker and therapeutic target for BC and TNBC. 

### 3.2. Tumor-Suppressive circRNA as miRNA Sponges in BC

Some studies have shown that many tumor-suppressive circRNAs play significant roles in delaying tumor progression by upregulating the tumor-suppressive genes related to proliferation, apoptosis, invasion, and migration. Table 2 summarizes the current research on the roles of tumor-suppressor circRNA in the development of BC. Tumor-suppressive circRNAs were observably downregulated in BC and TNBC tissues and cells. The overexpression of tumor-suppressive circRNA can absorb tumor-promoting miRNAs, upregulate tumor suppressor genes to weaken the capacity of cancer cell proliferation and colony formation, accelerate programmed cell death (PCD) (such as apoptosis), and inhibit epithelial–mesenchymal transformation (EMT) [71,72]. As a result, epithelioid cancer cells gradually lose their polarity and tight intracellular junctions, acquire a mesenchymal phenotype and stem characteristics, and reduce their resistance to drugs for BC treatment. The low expression of circRNA is negatively correlated with distant metastasis and the lethality of invasive BC. Low expression is also closely related to a larger tumor size, advanced TNM staging, lymph node metastasis, and a poor prognosis.

Many studies demonstrated that tumor-suppressive circRNAs negatively regulate the malignant biological behavior of BC and TNBC. For instance, nuclear receptor subfamily 3 group C member 2 (*NR3C2*) offers anti-metastatic activity and upregulates the expression of E3 ubiquitin ligase (*HRD1*) by sponging miR-513a-3p to target vimentin in TNBC, which plays a pivotal role in the EMT process by increasing cellular invasion and migration [73,74,75]. Ye et al. showed that circ_0001451 (circ_FBXW7) is derived from exons 3 and 4 of F-box and WD repeat domain containing 7 (*FBXW7*)—located on chromosome 4q31.3 (chr4:153332454–153333681)—through the back-splicing of exons 3 and 4. Circ_FBXW7 acts as a sponge to adsorb miR-197-3p and encodes the FBXW7-185aa protein, thereby inhibiting the proliferation and migration of TNBC cells by upregulating tumor-suppressor *FBXW7*, which induces c-Myc degradation [76]. Tamoxifen (TAM) is the most common endocrine therapy for patients with hormone receptor (HR)-positive breast cancer. Many patients eventually develop drug resistance after long-term clinic treatment, which remains a serious challenge [77]. Sang et al. reported that circ_0025202 with a length of 495 bp was spliced from glyceraldehyde-3-phosphate dehydrogenase (*GAPDH*) with a length of 688 bp. Circ_0025202 acts as an important regulator of tumor-suppressor and TAM-treatment sensitivity in HR-positive breast cancer cells by sponging miR-182-5p and inhibiting the 3′-untranslated region (UTR) binding of transcription factor forkhead box O3A (*FOXO3a*), thus inhibiting cell proliferation, colony formation, and migration and increasing cell apoptosis [77]. Many circRNAs also affect malignant phenotypes, such as circ_NOL10 [78], circ_TADA2As [79], circ_KDM4C [80], circ_0001283 [81], circ_DDX17 [82], circ_000554 [83], and circ_NFIC [84] (Table 2), by regulating the process of aerobic glycolysis and blocking the acquisition of ATP by cancer cells.

In summary, the expression of tumor-suppressor circRNA is mainly downregulated and acts as a sponge to absorb miRNA in the malignant progression of BC and TNBC, which relieves the inhibition effect of miRNA on the expression of the tumor-suppressor protein and limits the malignant growth of cancer cells, as well as the ability to engage in EMT. Meanwhile, circRNA promotes PCD and controls energy metabolism. The above-mentioned tumor-suppressor circRNA can be used as a target for the diagnosis and treatment markers of BC in the future and offers the possibility to reduce the resistance of patients to chemotherapy, as well as reduce tumor recurrence.

**Table 2 biomedicines-09-01590-t002:** The sponge functions of tumor-suppressor circRNA in BC and TNBC.

circBase ID (Alias)	microRNA	Direct/Indirect Targets	Phenotype	Reference
circ_NOL10	miR-767-5p	*SOCS2*	Proliferation, migration, invasion, EMT, tumor growth (BC)	[78]
circ_KDM4C	miR-548p	*PBLD*	Proliferation, migration, drug resistance (BC)	[80]
circ_0001283	miR-187	*HIPK3*	Cell growth, invasion (BC)	[81]
circ_DDX17	miR-605	*CDK1,p21*	Multiplication, colony formation (BC)	[82]
circ_000554	miR-182	*ZFP36*	EMT, invasion, migration (BC)	[83]
circ_NFIC	miR-658	*UPK1A*	Multiplication, migration (BC)	[84]
circ_0000442	miR-148b-3p	*PTEN*	Proliferation, tumor growth (BC)	[85]
circ_0000511	miR-326	*TAZ*	Proliferation, migration, invasion, apoptosis (BC)	[86]
circ_KLHL24	miR-1204	*ALX4*	Cell viability, colony formation, migration, invasion, glycolysis (BC)	[87]
circ_0025202	miR-182-5p	*FOXO3a*	Proliferation, colony formation, migration, apoptosis, drug resistance (BC)	[77]
circ_NR3C2	miR-513a-3p	*HRD1*	Proliferation, migration, invasion, EMT, ubiquitination (TNBC)	[73]
circ_TADA2As	miR-203a-3p	*SOCS3*	Proliferation, migration, invasion, clone formation (TNBC)	[79]
circ_FBXW7	miR-197-3p	*FBXW7*	Proliferation, migration, tumor growth (TNBC)	[76]

Abbreviations: SOCS2 (Suppressor Of Cytokine Signaling 2), PBLD (Phenazine Biosynthesis-Like Protein), HIPK3 (Homeodomain Interacting Protein Kinase 3), CDK1 (Cyclin Dependent Kinase 1), P21 (P21 Calcium Binding Protein), ZFP36 (ZFP36 Ring Finger Protein), UPK1A (Uroplakin 1A), PTEN (Phosphatase and Tensin Homolog), TAZ (Tafazzin, Phospholipid-Lysophospholipid Transacylase), ALX4 (ALX Homeobox 4), FOXO3a (Forkhead Box O3A), HRD1 (E3 Ubiquitin-Protein Ligase HRD1), SOCS3 (Suppressor Of Cytokine Signaling 3), FBXW7 (F-Box And WD Repeat Domain Containing 7). BC and TNBC refer to the role that circRNA plays in BC and TNBC, respectively.

## 4. The Dual Role of circRNA as miRNA Sponges in CRC

The expression of circRNA in CRC is also different than that in BC depending on the specific context. Hence, circRNA also plays a dual role in CRC during CRC initiation and development. Many studies showed that circRNA usually acted as a sponge by adsorbing miRNAs, which affects the expression of target genes by restoring their original functions in CRC (cancer-promoting or anti-tumor functions) through regulating the growth and development of the related proteins [88].

### 4.1. Oncogenic circRNA as miRNA Sponges in CRC

Numerous circRNAs are highly expressed in CRC cells and tissues, which bind multiple miRNAs through their multiple sites to upregulate target proteins; promote the growth, migration, and invasion of CRC cells; increase resistance to chemotherapy and the formation of stem cells [89]; or inhibit cell apoptosis (Table 3).

Various circRNAs act as oncogenic genes to promote growth and proliferation in CRC. The expression of circ_0055625, which is associated with pathological TNM staging and promoted tumor growth and metastasis, was found to be significantly upregulated in CRC tissues through the activation of integrin subunit beta 8 (*ITGB8*) via the sponging of miR-106b [90]. Another circRNA, circ_000166, was observably increased in human CRC tissues and cell lines. CRC cell viability, colony formation, migration, and invasion significantly decreased in vitro by downregulating the ETS transcription factor ELK1 (*ELK1*) through sponging miR-330-5p by circ_000166, in addition to decreased tumor size and weight in vivo [91]. The expression of circ_CAMSAP1, originating from exon 2 to exon 3 of calmodulin regulated spectrin associated protein 1 (*CAMSAP1*), is also significantly upregulated in CRC, which was significantly correlated with an advanced stage of TNM and shortened overall survival. Circ_CAMSAP1 acts as a sponge for miR-328-5p and eliminates miR-328-5p’s inhibition of the transcription factor *E2F1*, leading to the proliferation of CRC tumors [92]. Another circRNA, circ_100290, was positively correlated with tumor metastasis and negatively correlated with prognosis [93]. Circ_100290 is a ceRNA of frizzled class receptor 4 (*FZD4*) that leads to the activation of the Wnt/β-catenin pathway by absorbing miR-516b, which improves the proliferation of CRC. In addition, circ_CTNNA1 [94], circ_0038646 [95], and circ_0128846 [96] also promote the proliferation of CRC tumors.

Many oncogenic circRNAs are also involved in CRC cell migration and invasion. Li et al. [97] showed that circ_TBL1XR1 upregulated the expression of SMAD family member 7 (*Smad7*) by inhibiting miR-424, thereby facilitating the migration and invasion of CRC. Similarly, the overexpression of circ_0007843 acted as a sponge for miR-518C-5p and removed the inhibition of matrix metallopeptidase 2 (*MMP2*) transcription and translation, thereby increasing the invasion and migration in SW480 cells [98]. Circ_PPP1R12A, which is highly expressed in CRC, absorbs miR-375 and decreases the inhibition of miR-375 on catenin beta 1 (*CTNNB1*), thus increasing the invasion of CRC [94]. In addition, CRC patients with high circ_0001178 levels are more likely to display metastatic clinical features, advanced TNM stages, and adverse prognoses. Circ_0001178 suppresses miR-382/587/616 and upregulates *ZEB1* to EMT, thereby promoting metastatic dissemination in CRC. *ZEB1* also increases the expression of circ_0001178 by binding the promoter region of circ_0001178 [99]. Other circRNAs, such as circ_FARSA [100], have similar functions.

Masses of oncogenic circRNAs widely participate in drug resistance. Circ_0055625 works as a sponge for miR-338-3p to regulate the expression of Musashi RNA-binding protein 1 (*MSI1*) in CRC cells. The knockout of circ_0055625 inhibits tumor growth and improves radiosensitivity in vivo [101]. Further, circ_0079662 acts as a ceRNA to bind miR-324-5p, which regulates target gene homeobox A9 (*HOXA9*) through the tumor necrosis factor alpha (*TNF-α*) pathway and induces resistance to the chemotherapy drug oxaliplatin in CRC [102]. Radiation therapy in colon cells induces significant upregulation in the expression of circ_0001313, as well as the downregulation of miR-338-3p, thus increasing CRC tumor radioresistance and suggesting a positive correlation between CRC radioresistance and circ_0001313 [103]. Other circRNAs, such as circ_0020095 [104], circ_0005963 [101], and circ_0000338 [105], have similar functions as oncogenics involved in CRC tumors.

In addition, multiple oncogenic circRNAs take part in the processes of cell cycle and cell apoptosis. Zhou et al. [106] reported that circRNA_100859 upregulates the expression of *HIF1A* by negatively regulating the expression of miR-217, which inhibits cell apoptosis in CRC tumors. Circ_0005615 acts as a carcinogenic gene in CRC by sponging miR-149-5p to release tankyrase (*TNKS*) by activating the Wnt/β-catenin signal pathway and upregulating the expression of cyclin D1 (*CCND1*), thus accelerating the CRC cell cycle [107]. As a sponge of miR-874-3p, the overexpression of circ_0007142 in CRC prevents the inhibition of miR-874-3p on the target glycerophosphodiester phosphodiesterase domain containing 5 (*GDPD5*), thereby promoting the apoptosis of CRC [108]. Some other circRNAs, such as circ_0055625 [101] and circ_0006990 [109], also play the same roles in CRC cells.

Some oncogenic circRNAs are involved in the stem cell activity, angiogenesis, and metabolism in CRC. The expression of circ_0001806 was upregulated in CRC tissues and involved in TNM staging, invasion, lymphatic metastasis, and distant metastasis. Circ_0001806 promotes the phenotype of CRC stem cells by activating the circ_0001806/miR-193-5p/collagen type I alpha 1 chain (*COL1A1*) axis [110]. On the other hand, to grow rapidly and resist chemotherapy, many malignant tumors, including CRC, can frequently rely on aerobic glycolysis to produce ATP. Circ_0005963, a sponge of miR-122 that targets pyruvate kinase M2 (*PKM2*), is positively correlated with chemical resistance. Studies both in vitro and in vivo have shown that exosomes from oxaliplatin-resistant cells deliver circ_0005963 to sensitive cells, thus increasing glycolysis and drug resistance through sponging miR-122 and upregulating the expression of *PKM2* [111]. Moreover, Guo et al. [112] reported that circ_3823 works as a ceRNA of miR-30c-5p, which alleviates the inhibition of miR-30c-5p on its target transcription factor 7 (*TCF7*) in vivo and vitro, which is involved in promoting the growth, metastasis, and angiogenesis of CRC.

**Table 3 biomedicines-09-01590-t003:** The sponge functions of cancer-promoting circRNA in CRC.

circBase ID (Alias)	microRNA	Direct/Indirect Targets	Phenotype	References
circ_100859	miR-217	*HIF1A*	Tumor growth, apoptosis	[106]
circ_0055625	miR-106b	*ITGB8*	Proliferation, migration, invasion	[90]
circ_ACAP2	miR-21-5p	*Tiam1*	Proliferation, migration, invasion	[113]
circ_000166	miR-330-5p	*ELK1*	Tumor growth, colony formation, migration, invasion	[91]
circ_100290	miR-516b	*FZD4*	Proliferation, migration	[93]
circ_TBL1XR1	miR-424	*Smad7*	Migration, invasion	[97]
circ_0007843	miR-518c-5p	*MMP2*	Invasion, migration	[98]
circ_0001313	miR-338-3p	unclear	Radiation resistance	[103]
circ_0055625	miR-338-3p	*MSI1*	Radio resistance, proliferation, migration, invasion, apoptosis	[101]
circ_CTNNA1	miR-149-5p	*FOXM1*	Proliferation, invasion	[94]
circ_PPP1R12A	miR-375	*CTNNB1*	Proliferation, invasion	[114]
circ_0008274	miR-140-3p	*GRN*	Proliferation, migration, invasion	[115]
circ_0079662	mir-324-5p	*HOXA9*	Drug resistance	[102]
circ_CAMSAP1	miR-328-5p	*E2F1*	Proliferation, invasion	[92]
circ_0020095	miR-487a-3p	*SOX9*	Proliferation, migration, invasion, drug resistance	[104]
circ_0038646	miR-331-3p	*GRIK3*	Proliferation, migration	[95]
circ_0001178	miR-382/587/616	*ZEB1*	Migration, invasion	[99]
circ_0001806	miR-193-5p	*COL1A1*	Stem cells	[110]
circ_0128846	miR-1184	*AJUBA*	Proliferation, migration	[96]
circ_0006990	miR-101-3p	unclear	Proliferation, migration, invasion, apoptosis	[109]
circ_0005963	miR-122	*PKM2*	Glycolysis, drug resistance	[111]
circ_AGFG1	miR-4262 miR-185-5p	*YY1CTNNB1*	Migration, stem cell	[116]
circ_FARSA	miR-330-5p	*LASP1*	Proliferation, migration, invasion	[100]
circ_HIPK3	miR-7	*FAK/IGF1R/EGFR/YY1*	Tumor growth, migration	[117]
circ_0005615	miR-149-5p	*TNKS*	Proliferation, cell cycle	[107]
circ_0000218	miR-139-3p	*RAB1A*	Proliferation, migration	[118]
circ_0060745	miR-4736	*CSE1L*	Proliferation, migration	[119]
circ_3823	miR-30c-5p	*TCF7*	Proliferation, migration, angiogenesis	[112]
circ_0000338	miR-217/miR-485-3p	unclear	Drug resistance	[105]
circ_0007142	miR-874-3p	*GDPD5*	Proliferation, apoptosis	[108]

Abbreviations: HIF-1α (Hypoxia-Inducible Factor 1 subunit alpha), ITGB8 (Integrin Subunit Beta 8), Tiam1 (TIAM Rac1 Associated GEF 1), ELK1 (ETS Transcription Factor ELK1), FZD4 (Frizzled Class Receptor 4), Smad7 (SMAD Family Member 7), MMP2 (Matrix Metallopeptidase 2), MSI1 (Musashi RNA Binding Protein 1), FOXM1 (Forkhead Box M1), CTNNB1 (Catenin Beta 1), GRN (Granulin Precursor), HOXA9 (Homeobox A9), E2F1 (E2F Transcription Factor 1), SOX9 (SRY-Box Transcription Factor 9), GRIK3 (Glutamate Ionotropic Receptor Kainate Type Subunit 3), ZEB1 (Zinc Finger E-box Binding Homeobox 1), COL1A1 (Collagen Type I Alpha 1 Chain), AJUBA (Ajuba LIM protein), PKM2 (Pyruvate Kinase M2), YY1 (YY1 Transcription Factor), LASP1 (LIM And SH3 Protein 1), FAK (Focal Adhesion Kinase), IGF1R (Insulin Like Growth Factor 1 Receptor), EGFR (Epidermal Growth Factor Receptor), TNKS (Tankyrase), RAB1A (Member RAS Oncogene Family), CSE1L (Chromosome Segregation 1 Like), TCF7 (Transcription Factor 7), GDPD5 (Glycerophosphodiester Phosphodiesterase Domain Containing 5).

### 4.2. Tumor-Suppressive circRNA as miRNA Sponges in CRC

Similar to tumor-suppressor genes, many downregulated tumor-suppressive circRNAs exist in CRC, which means that miRNAs are not restricted, and target genes are suppressed in this way. The overexpression of these circRNAs downregulates the expression of miRNAs and activates tumor-suppressive genes (Table 4).

Many tumor-suppressive circRNAs are widely involved in CRC cell growth and proliferation. Circ_103809 was proven to be expressed at low levels in CRC tissues and relates to cancer tissue staging and lymph node metastasis. The knockout of circ_103809 enhances the proliferation and migration of CRC cells. Mechanistically, circ_103809 downregulates the expression of miR-532-3p, thereby upregulating the expression of forkhead box O4 (*FOXO4*), which relates to the proliferation of CRC [120]. Similarly, Li et al. [121] revealed that the expression of circ_CBL.11 significantly increases in CRC cells through carbon ion irradiation. Circ_CBL.11 regulates the expression of tyrosine 3-monooxygenase/tryptophan 5-monooxygenase activation protein epsilon (*YWHAE*) by sponging miR-6778-5p to directly suppress cell proliferation in CRC, which plays an important role in improving the efficacy of carbon ions against CRC. Circ_0009361 is downregulated in CRC tissues and cells and acts as a sponge for miR-582. Circ_0009361 upregulates the expression of the target protein adenomatous polyposis coli 2 (*APC2*) and inhibits the Wnt/β-catenin signaling pathway, which exerts a tumor-suppressive effect on the growth, invasion, and metastasis in CRC [122]. Other circRNAs, such as circ_CDYL [123], have similar functions in the development of CRC.

Moreover, many studies indicated that multiple tumor-suppressive circRNAs take part in the migration and invasion of CRC cells. Low-expression of circ_0008285 can upregulate the expression of *PTEN* through sponging miR-382-5p in CRC, thereby, decreasing the proliferation and migration of CRC cells by inhibiting the *PI3K*/*AKT* pathway [124]. Circ_SMARCA5 modulates the development of cancers through inhibiting migration and invasion, which involves blocking the Wnt and Yes1 associated transcriptional regulator (*YAP1*) pathways by blocking miR-552 in CRC [125]. In addition, circ_0026344 [126] also has a similar function to Circ_FNDC3B in CRC.

Furthermore, some circRNAs have tumor-suppressive functions in the progress of the cell cycle and apoptosis in CRC. Circ_ITGA5 sponges miR-107, which upregulates the expression of forkhead box J3 (*FOXJ3*); highly expressed *FOXJ3* inhibits proliferation and migration and facilitates apoptosis in CRC cells [127]. Cui et al. reported that the expression of circ_CDYL generally decreased in CRC tissues, while the expression of miR-150-5p was elevated in both normal tissues and CRC. Circ_CDYL reduces cell viability and promotes apoptosis by downregulating *c-Myc* and cyclin D1 (*CCND1*) and upregulating *p53*, cleaved caspase-3 (*Casp3*), and poly (ADP-ribose) polymerase (*PARP*) in SW480 and SW620 cells [123]. In addition, the expression of circ_0026344 is downregulated in CRC tissues. Circ_0026344 can inhibit tumor growth and invasion and promote apoptosis, suggesting that the upregulation of circ_0026344 is beneficial to the treatment of CRC [126]. Similar results were found for circ_104916 [128].

Taken together, the oncogenic and tumor-suppressive functions of circRNAs in CRC have been widely reported. Overall, circRNA acts as a miRNA sponge involved in proliferation, migration, invasion, drug resistance, and apoptosis. More circRNAs that promote or inhibit the growth of CRC are expected to be identified in the future.

**Table 4 biomedicines-09-01590-t004:** The sponge function of tumor-suppressor circRNA in CRC.

circBase ID (Alias)	microRNA	Direct/Indirect Targets	Phenotype	References
circ_103809	miR-532-3P	*FOXO4*	Proliferation, migration	[120]
circ_ITGA5	miR-107	*FOXJ3*	Proliferation, migration, apoptosis	[127]
circ_CBL.11	miR-6778-5p	*YWHAE*	Proliferation	[121]
circ_SMARCA5	miR-552	unclear	Tumor growth, migration, invasion	[113,125]
circ_CDYL	miR-105-5p	*c-Myc/cnd /p53/Casp3/PARP*	Tumor growth, apoptosis	[123]
circ_0026344	miR-21/31	unclear	Tumor growth, invasion, apoptosis	[126]
circ_0008285	miR-382-5p	*PTEN*	Proliferation, migration	[124]
circ_0009361	miR-582	*APC2*	Tumor growth, migration	[122]

Abbreviations: FOXO4 (Forkhead Box O4), FOXJ3 (Forkhead Box J3), YWHAE (Tyrosine 3-Monooxygenase/Tryptophan 5-Monooxygenase Activation Protein Epsilon), c-Myc (BHLH Transcription Factor), ccnd1 (CyclinD1), p53 (Tumor Protein P53), Casp3 (Caspase-3), PARP (Poly-(ADP-Ribose) Polymerase), PTEN (Phosphatase and Tensin Homolog), APC2 (APC Regulator of WNT Signaling Pathway 2).

## 5. Additional Roles of circRNA in BC and CRC and Sponge Functions in other Cancers

In addition to acting as miRNA sponges, circRNAs play other roles in BC and CRC. On the other hand, circRNAs also widely function as miRNA sponges in other cancers.

### 5.1. miRNA Sponge Function of circRNA in other Cancers 

When acting as a miRNA sponge, circRNA also regulates the progression of other cancers in terms of cell proliferation, apoptosis, migration, invasion, glycolysis, and drug resistance. Circ_0026123 potentially promotes the proliferation of ovarian cancer cells by activating the miR-124-3p/enhancer of zeste 2 polycomb repressive complex 2 subunit (*EZH2*) signaling [129]. The inhibition of circ_0000285 prevents cell proliferation and induces apoptosis in thyroid cancer by sponging miR-654-3p [130]. Circ_WHSC1 promotes proliferation, migration, and invasion and inhibits apoptosis in endometrial cancer by sponging miR-646 and targeting nucleophosmin 1 (*NPM1*) [131]. Tang et al. found that circ_SETD3 competitively adsorbs to miR-615-5p and miR-1538 and negates their inhibitory effects on microtubule-associated protein RP/EB family member 1 (*MAPRE1*) mRNA, thereby upregulating the expression of *MAPRE1* and enhancing the invasion and migration capabilities of nasopharyngeal carcinoma (NPC) cells [132]. Su et al. found that circ_RIP2 can sponge miR-1305 to elevate transforming growth factor beta 2 (*TGFB2*) in bladder cancer in addition to inducing EMT via the *TGFB2*/SMAD family member 3 (*SMAD3*) pathway [133]. Circ_PRKCI targets miR-1294 and miR-186-5p by downregulating forkhead box K1 (*FOXK1*) expression to suppress glycolysis in hepatocellular carcinoma [134]. Circ_EPHB4 upregulates the expression of SRY-box transcription factor 10 (*SOX10*) and Nestin (*NES*) by directly sponging miR-637, thereby stimulating the stemness, proliferation, and glycolysis of glioma cells [135]. Circ_ASAP1 promotes tumorigenesis and the temozolomide (TMZ) resistance of glioblastoma via the NRAS proto-oncogene, GTPase (*NRAS*)/mitogen-activated protein kinase kinase 1 (*MEK1*)/extracellular signal-regulated kinase 1 and 2 (*ERK1**-2*) signaling pathway [136]. 

### 5.2. Additional Roles of circRNA in BC and CRC

In addition to acting as miRNA sponges, current studies show that circRNAs play roles in BC and CRC in additional ways (see Figure 2 for details): (a) circRNAs can bind to host genes at their synthesis loci and cause transcriptional pausing or termination through the formation of an RNA–DNA hybrid (R-loop structure), thereby upregulating exon-skipped or truncated transcripts. Xu et al. demonstrated that circ_SMARCA5 can bind to its parent gene locus, thus forming an R-loop and resulting in transcriptional pausing at exon 15 of the SWI/SNF-related, matrix-associated, actin-dependent regulator of chromatin, subfamily a, member 5 (*SMARCA5*). Circ_SMARCA5 expression resulted in the downregulation of *SMARCA5* and the production of a truncated nonfunctional protein. Moreover, the overexpression of circ_SMARCA5 was sufficient to improve sensitivity to cytotoxic drugs [137]. Circ_SCRIB enhances BC progression by suppressing parental gene splicing and translation [138]. (b) EIciRNAs can combine with U1 snRNP and then interact with RNA polymerase II (Pol II) to enhance parental gene expression [139]; however, this regulation function requires further study in BC and CRC. (c) CircRNAs can also serve as sponges or decoys with proteins [140,141]. Chen et al. found that circ_RHOBTB3 binds to Hu-antigen R (HuR), which is a ubiquitously expressed and functional component of the RNA-binding protein (RBP) in CRC development and promotes the expression of the β-Trcp1-mediated ubiquitination HuR. HuR is a member of the ELAV family and an RBP [142] that post-transcriptionally modulates its target genes by stabilizing their mRNAs and is involved in cell growth and tumorigenesis [143,144]. In addition, blocking the interactions between circ_AGO2 and HuR using cell-penetrating inhibitory peptides represses the tumorigenesis and aggressiveness of CRC [145]. (d) Interestingly, circRNAs are predicted to include an open reading frame (ORF) with upstream internal ribosome entry sites (IRES), indicating that circRNA has the potential to be translated into proteins [146]. Moreover, the Cap-independent translation of circRNAs can occur through IRESs [147] or following the incorporation of m6A RNA modification at the 5′-UTR [148,149]; however, circRNAs lack the 5′cap and the poly(A) tail. Ivano et al. demonstrated that several endogenous circRNAs, such as circ_ZNF609, circ_Mbl, circ_FBXW7, circ_PINTexon2 and circ_SHPRH, serve as important protein templates [150,151,152,153,154,155,156]. However, the functional relevance of most circRNA-derived peptides in BC and CRC remains unknown.

## 6. Conclusions

In this review, we described the dual role of circRNA as miRNA sponges that either promote or suppress the progression of cell proliferation, migration, and invasion in BC and CRC. Moreover, some genes were found to be commonly involved in both BC and CRC regulation, such as *BRAF*, *PIK3CA*, *ERBB2*, *TP53*, *PTEN*, *FGFR1*, *ERBB3*, neuregulin 1 (*NRG1*), *GNAS*, *KDR*, and *PSMD4*, which are regulated by miR-338-3p, miR-375, and miR-149-5p. While each of these miRNAs can be targeted by two or more circRNAs in BC and CRC, miR-338-3p is targeted by circ_TFF1, circ_0001313, and circ_0055625; miR-375 by circ_KIF4A and circ_PPP1R12A; and miR-149-5p by circ_0072995 and circ_0005615. These miRNAs can also play their roles individually as miR-338-3p was observed to respond to 5-FU treatment in CRC through miR-338-3p/mTOR/autophagy in a p53-dependent manner [157]. MiR-338-3p was found to block the growth of gastric cancer cells through the phosphatase and *PTEN*/*PI3K* signaling pathways [158], and miR-375 inhibited CRC growth by targeting *PIK3CA* [159]. These findings suggest that circRNAs are widely involved in the initiation and development of BC and CRC through the same signal pathways. 

To date, most studies on the miRNA sponge functions of circRNAs in BC and CRC primarily described the indirect relationship between the circRNA–miRNA axis and malignant behaviors [160]. Further investigations on the upstream and downstream regulatory networks are required to improve the accuracy of circRNAs as diagnostic markers. Furthermore, the miRNA sponge functions of many circRNAs have been questioned, partially because circRNAs are far less abundant than miRNAs, and the majority of circRNAs harbor far fewer miRNA binding sites than the circRNAs ciRS-7 and circ_ZNF91 [39,161]. Therefore, the circRNA to miRNA ratios should be taken into account to obtain non-biased conclusions. In addition, distinguishing between the exonic types and intron and exon combination types of patterns in circRNA structures would rule out other functions of circRNAs beyond their roles as miRNA sponges. Further circRNA explorations could yield a new wave of diagnostic and therapeutic biomarkers that target BC and CRC based on their foundational pathophysiology rather than simply describing and suppressing their symptoms.

## Figures and Tables

**Figure 1 biomedicines-09-01590-f001:**
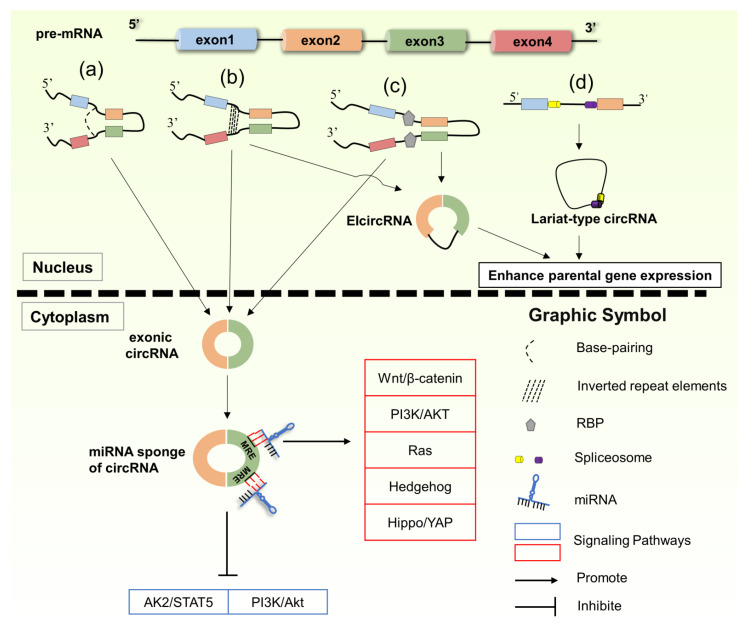
The formation of circRNA and its downstream target signal pathway in BC or CRC. (**a**) exonic circRNA; (**b**) circRNA combined with introns and exons, also known as elcircRNA; (**c**) circRNA formation is regulated by RNA-binding proteins (RBPs); (**d**) lariat-type circRNA composed of introns. Abbreviations: MRE (miRNA response element).

**Figure 2 biomedicines-09-01590-f002:**
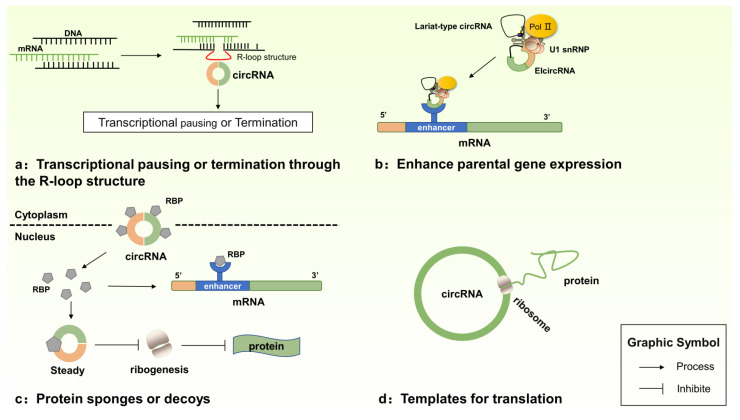
The other functions of circRNAs. (**a**) Transcriptional pausing or termination through the R-loop structure; (**b**) enhance parental gene expression; (**c**) protein sponges or decoys; (**d**) templates for translation. Abbreviations: U1 snRNP (U1 small nuclear ribonucleoprotein), RBP (RNA-binding proteins).

**Table 1 biomedicines-09-01590-t001:** The sponge function of cancer-promoting circRNA in BC and TNBC.

circBase ID (Alias)	microRNA	Direct/Indirect Targets	Phenotype	Reference
circ_0000518	miR-326	*FGFR1*	Cell cycle, proliferation, colony formation, migration and invasion, tumor growth (BC)	[43]
circ_MMP11	miR-1204	*MMP11*	Multiplication, migration (BC)	[57]
circ_002178	miR-328-3p	*COL1A1*	Cell viability, energy metabolism, blood vessel formation (BC)	[68]
circ_0000291	miR-326	*ETS1*	Proliferation, migration, invasion (BC)	[58]
circ_YY1	miR-769-3p	*YY1*	Cell viability, colony formation, migration, invasion, glycolysis (BC)	[66]
circ_HIPK3	miR-193a	*HMGB1*	Proliferation, migration, invasion (BC)	[50]
circ_TFF1	miR-338-3p	*FGFR1*	Proliferation, invasion, glycolysis, apoptosis (BC)	[46]
circ_DCAF6	miR-616-3p	*GLI1*	Cell growth, stem cell stemness (BC)	[67]
circ_PLK1	miR-4500	*IGF1*	Cell growth, migration, invasion (BC)	[48]
circ_0011946	miR-26a/b	*RFC3*	Migration (BC)	[56]
circ_0007534	miR-593	*MUC19*	Proliferation, colony formation, invasion, apoptosis (BC)	[47]
circ_WWC3	miR-26b-3p miR-660-3p	*ZEB1*	Proliferation, migration, invasion (BC)	[55]
circ_MYO9B	miR-4316	*FOXP4*	Proliferation, migration, invasion, tumor growth (BC)	[59]
circ_ZFR	miR-578	*HIF1A*	Cell viability, colony formation, migration, invasion, glycolysis, apoptosis (BC)	[64]
circ_ABCC4	miR-154-5p	*NF-1*	Cell viability, migration, invasion (BC)	[45]
circ_0072995	miR-149-5p	*SHMT2*	Anaerobic glycolysis (BC)	[65]
circ_0008039	miR-432-5p	*E2F3*	Proliferation, migration, cell cycle (BC)	[44]
circ_TP63	miR-873-3p	*FOXM1*	Proliferation, cell cycle, invasion, migration, tumor growth (BC)	[49]
circ_IFI30	miR-520b-3p	*CD44*	Proliferation, migration, invasion, cell cycle, apoptosis (TNBC)	[51]
circ_UBE2D2	miR-512-3p	*CDCA3*	Proliferation, migration, invasion, tumor growth, drug resistance (TNBC)	[69]
circ_KIF4A	miR-375	*KIF4A*	Multiplication, migration (TNBC)	[70]
circ_GNB1	miR-141-5p	*IGF1R*	Proliferation, migration, tumor growth (TNBC)	[42]
circ_BACH2	miR-186-5p/miR-548c-3p	*CXCR4*	Proliferation, EMT (TNBC)	[54]
circ_RAD18	miR-208a/miR-3164	*IGF1/FGF2*	Proliferation, migration, apoptosis, tumor growth (TNBC)	[60]

Abbreviations: FGFR1 (Fibroblast Growth Factor Receptor 1), MMP11 (Matrix Metallopeptidase 11), COL1A1 (Collagen Type I Alpha 1 Chain), ETS1 (ETS Proto-Oncogene 1), YY1 (YY1 Transcription Factor), HMGB1 (High Mobility Group Box 1), FGFR1 (Fibroblast Growth Factor Receptor 1), GLI1 (GLI Family Zinc Finger 1), IGF1 (Insulin-Like Growth Factor 1), RFC3 (Replication Factor C Subunit 3), MUC19 (Mucin 19, Oligomeric), ZEB1 (Zinc Finger E-box Binding Homeobox 1), FOXP4 (Forkhead Box P4), HIF1A (Hypoxia Inducible Factor 1 Subunit Alpha), NF-1 (Neurofibromin 1), SHMT2 (Serine Hydroxymethyltransferase 2), E2F3 (E2F transcription factor 3), FOXM1 (Forkhead Box M1), CD44 (CD44 Molecule), CDCA3 (Cell Division Cycle Associated 3), KIF4A (Kinesin Family Member 4A), IGF1R (Insulin-Like Growth Factor 1 Receptor), CXCR4 (C-X-C Motif Chemokine Receptor 4), FGF2 (Fibroblast Growth Factor 2). BC and TNBC refer to the role that circRNA plays in BC and TNBC, respectively.

## Data Availability

Not applicable.

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
