# Peer review of "The Dual Role of Circular RNAs as miRNA Sponges in Breast Cancer and Colon Cancer"

_biomedicines, 2021, doi:10.3390/biomedicines9111590_

Round 1

Reviewer 1 Report

The authors need to improve on the English language grammar and spelling used in the manuscript for acceptance to publish.

Author Response

Response to Reviewer 1 Comments

Point 1: Comments and Suggestions for Authors

The authors need to improve on the English language grammar and spelling used in the manuscript for acceptance to publish.

Response 1: We have already extensively revised our English language grammar and spelling through the language editing service offered by MDPI in the revised manuscript.

Reviewer 2 Report

I am sorry for my delay in response.

The manuscript submitted is a review article of circular RNA as miRNA sponges in breast cancer and colon cancer.  Though there are many similar review articles, of which some are more intensive and in detail, for breast cancer, there are few similar review articles for colorectal cancers, this might be of some significance.  In addition, they clearly divided circRNAs in view of onogenic vs tumor suppressor, which is interesting. However, the merit of this review do not appear so large in comparison with previously published review articles handling the same subject.  Specific comments are as follows:

  1. The terms absorb and adsorb, sometimes swallow, appear to be used in the same meaning. In English, do they have the same meaning?
  2. The number of References should be in the order of appearance in text. However, the authors describe the number of References in the order of appearance in Tables.
  3. The abbreviations should be arranged alphabetically.
  4. In Table 1 and page 10, line 360, reference [48] might be erroneous since the contents are different between the table and title of the reference.
  5. In Table 1 and page 10, line 360, reference [48] might be erroneous since the contents are different between the table and title of the reference.
  6. In page 7, line 256, reference [70] might be erroneous since the contents are different between the table and title of the reference.
  7. Prefix “hsa” is frequently described as “has”.
  8. In page 1, line 32-34, are the contents of sentence “TNBC contributes to more than 90% of deaths due to high metastasis” accurate?
  9. In page 8, line 278-279, are the contents of sentence “unlike to the BC or TNBC, the studies of circRNA were used as the diagnosis …” accurate? This sentence appear to indicate that circRNA is already used in the diagnosis of BC or TNBC.
  10. The format of References are incomplete.

Author Response

Response to Reviewer 2 Comments

Comments and Suggestions for Authors

The manuscript submitted is a review article of circular RNA as miRNA sponges in breast cancer and colon cancer.  Though there are many similar review articles, of which some are more intensive and in detail, for breast cancer, there are few similar review articles for colorectal cancers, this might be of some significance.  In addition, they clearly divided circRNAs in view of onogenic vs tumor suppressor, which is interesting. However, the merit of this review do not appear so large in comparison with previously published review articles handling the same subject.  Specific comments are as follows:

Point 1: The terms absorb and adsorb, sometimes swallow, appear to be used in the same meaning. In English, do they have the same meaning?

Response 1: We have corrected and changed the word "swallow" to "absorb" in our revised manuscript (page 8).

Point 2: The number of References should be in the order of appearance in text. However, the authors describe the number of References in the order of appearance in Tables.

Response 2: We have already rearranged the number of References in the order of appearance in the revised manuscript.

Point 3: The abbreviations should be arranged alphabetically.

Response 3: We have already arranged the abbreviations (gene/proteins abbreviations excluded) alphabetically in the end of our revised manuscript. Due to there were masses of gene/proteins abbreviations in this paper, and we have already gave their full name when they appeared firstly in this study, so, we did not arranged the gene/proteins abbreviations alphabetically.

Point 4: In Table 1 and page 10, line 360, reference [48] might be erroneous since the contents are different between the table and title of the reference.

Response 4: We have already corrected the position of this reference in the revised manuscript.

Point 5: In page 7, line 256, reference [70] might be erroneous since the contents are different between the table and title of the reference.

Response 5: We have already corrected the position of this reference in the revised manuscript.

Point 6: Prefix “hsa” is frequently described as “has”.

Response 6: To uniformly description the name of circRNA in the whole manuscript, we deleted the prefix ‘hsa’ in the revised manuscript due to all the circRNA in this review appeared in homo sapiens.

Point 7: In page 1, line 32-34, are the contents of sentence “TNBC contributes to more than 90% of deaths due to high metastasis” accurate?

Response 7: We have corrected this sentence to “Triple-negative breast cancer (TNBC, ER, PR, and HER2 are negatively expressed) contributes the high mortality of BC patients since it is closely associated with the high level of metastasis.” in the revised manuscript.

Point 8: In page 8, line 278-279, are the contents of sentence “unlike to the BC or TNBC, the studies of circRNA were used as the diagnosis …” accurate? This sentence appear to indicate that circRNA is already used in the diagnosis of BC or TNBC.

Response 8: The description is inaccurate and we have deleted this sentence from the revised manuscript due to many studies gave the expectation that circRNAs could be used as a diagnostic biomarker, however, very few or even no circRNAs were used in the clinical diagnosis of BC or TNBC, such as:

i). “Wan, L., Q. Han, B. Zhu, Z. Kong, and E. Feng."Circ-Tff1 Facilitates Breast Cancer Development Via Regulation of Mir-338-3p/Fgfr1 Axis." Biochem Genet (2021). DOI: 10.1007/s10528-021-10102-6”,

ii). “Liang, Y., X. Song, Y. Li, P. Su, D. Han, T. Ma, R. Guo, B. Chen, W. Zhao, Y. Sang, N. Zhang, X. Li, H. Zhang, Y. Liu, Y. Duan, L. Wang, and Q. Yang. "Circkdm4c Suppresses Tumor Progression and Attenuates Doxorubicin Resistance by Regulating Mir-548p/Pbld Axis in Breast Cancer." Oncogene 38, no. 42 (2019): 6850-66. DOI: 10.1038/s41388-019-0926-z”,

iii). “Zhang, J., H. Liu, P. Zhao, H. Zhou, and T. Mao. "Has_Circ_0055625 from Circrna Profile Increases Colon Cancer Cell Growth by Sponging Mir-106b-5p." J Cell Biochem 120, no. 3 (2019): 3027-37. DOI: 10.1002/jcb.27355”. et al.

Point 9: The format of References are incomplete.

Response 9: We have already checked and corrected the format of references in the revised manuscript.

Reviewer 3 Report

This review article adds to a growing list of circRNA reviews in the current literature. Although the authors focus on the area of CRC and BC research looking at miRNA sponging mechanism of circRNA over the past 3 years, there are still important findings at the earlier years that, in my opinion, should be discussed. The Tables produced in this manuscript are very useful for readers to have a quick glance on the interaction between the circRNA and miRNAs and the reported phenotypes. These Tables can be significantly improved with my comments below. Overall, this is a unique review article and can be of interests to researchers and clinicians looking at particular cancers with a specific angle.

Minor comments:

  • Replace the word absorb in the manuscript with sponge.
  • Gene/protein nomenclature needs to be properly applied to the manuscript. E.g. human genes to be (e.g. IGF1) and proteins non italicized.
  • It would be informative to indicate the number of deaths with CRC since this was reported with BC.
  • References needed for sentences in Line 77 – 81.
  • Line 88: replace All-exon with exonic. Biogenesis of exonic circRNAs can be formed via complementary introns flanking.
  • Sentence at Line 96 does not make sense. Need rephrasing.
  • References lacking in Line 106-108.
  • References needed in Line 446-448.
  • Sentence in Line 151 needs rephrasing.
  • Change MiR-148b-3p to miR-148b-3p in Table 2.
  • Reference 84 should be excluded from Table since the authors only include studies based in the last 3 years.

Major comments:

  • The English of this manuscript can be significantly enhanced with additional proofreading.
  • The authors need to highlight whether the reported literature are clinical, in vivo, or in vitro studies. It will be useful to include them in the Tables shown.
  • There are circRNAs that can undergo translation. This needs to be discussed and highlight any articles that are involved in these cancers.
  • Also, it will be informative to include if the circRNAs are upregulated or downregulated in the reported research, if any, into the Tables.
  • Figure 1 needs more work. The cartoons are not easily understood, and legends are not self-sufficient. The authors need to identify the different cartoons shown. E.g splice sites, blue pacman, etc. illustration needs to be clearer with lesser colors used.
  • The authors need to emphasis or suggest various methods on leveraging circRNAs in the treatment or prognosis of BC and CRC. For example, there has been a few research working on artificial circRNAs to sponge specific miRNAs or proteins. These suggestions can be, if possible, included in Figure 1, to show readers on the different pathways that can be exploited to treat or prognose the disease.
  • There are some biomarkers that used circRNA for prognosis. This information is useful to be included.

Author Response

Response to Reviewer 3 Comments

Comments and Suggestions for Authors

This review article adds to a growing list of circRNA reviews in the current literature. Although the authors focus on the area of CRC and BC research looking at miRNA sponging mechanism of circRNA over the past 3 years, there are still important findings at the earlier years that, in my opinion, should be discussed. The Tables produced in this manuscript are very useful for readers to have a quick glance on the interaction between the circRNA and miRNAs and the reported phenotypes. These Tables can be significantly improved with my comments below. Overall, this is a unique review article and can be of interests to researchers and clinicians looking at particular cancers with a specific angle.

Minor comments:

Point 1: Replace the word absorb in the manuscript with sponge.

Response 1: We retained the description of "absorb" in the revised manuscript due to the word "absorb" was widely used to descript that the circRNAs were used as miRNA sponging in many studies.

Point 2: Gene/protein nomenclature needs to be properly applied to the manuscript. E.g. human genes to be (e.g. IGF1) and proteins non italicized.

Response 2: We have corrected the formats of gene/protein nomenclature (the genes were italicized and the proteins were non italicized) in the revised manuscript.

Point 3: It would be informative to indicate the number of deaths with CRC since this was reported with BC.

Response 3: According the reference: "Siegel, R. L., K. D. Miller, A. Goding Sauer, S. A. Fedewa, L. F. Butterly, J. C. Anderson, A. Cercek, R. A. Smith, and A. Jemal. "Colorectal Cancer Statistics, 2020." CA Cancer J Clin 70, no. 3 (2020): 145-64", DOI: 10.3322/caac.21601, approximately 147,950 individuals were diagnosed with CRC and 53,200 died from the disease in 2020 in the United States, and we added the related description in the revised manuscript.

Point 4: References needed for sentences in Line 77 – 81.

Response 4: We have added the following references in the revised manuscript:

  1. i) "Zhang, H. D., L. H. Jiang, D. W. Sun, J. C. Hou, and Z. L. Ji. "Circrna: A Novel Type of Biomarker for Cancer." Breast Cancer 25, no. 1 (2018): 1-7"; DOI: 10.1007/s12282-017-0793-9.
  2. ii) "Qu, S., X. Yang, X. Li, J. Wang, Y. Gao, R. Shang, W. Sun, K. Dou, and H. Li. "Circular Rna: A New Star of Noncoding Rnas." Cancer Lett 365, no. 2 (2015): 141-8"; DOI: 10.1016/j.canlet.2015.06.003.

Point 5: Line 88: replace All-exon with exonic. Biogenesis of exonic circRNAs can be formed via complementary introns flanking.

Response 5: We have replaced the word "all-exon" with "exonic" in the revised manuscript

Point 6: Sentence at Line 96 does not make sense. Need rephrasing.

Response 6: We have removed this sentence in our revised manuscript.

Point 7: References lacking in Line 106-108.

Response 7: We have added the following references in the revised manuscript.

  1. i) "Anastasiadou, E., A. Faggioni, P. Trivedi, and F. J. Slack. "The Nefarious Nexus of Noncoding Rnas in Cancer." Int J Mol Sci 19, no. 7 (2018) "; DOI: 10.3390/ijms19072072.
  2. ii) "Tay, Y., J. Rinn, and P. P. Pandolfi. "The Multilayered Complexity of Cerna Crosstalk and Competition." Nature 505, no. 7483 (2014): 344-52"; DOI: 10.1038/nature12986.

Point 8: References needed in Line 446-448.

Response 8: We have added the following reference in the revised manuscript.

"Li, Z., C. Huang, C. Bao, L. Chen, M. Lin, X. Wang, G. Zhong, B. Yu, W. Hu, L. Dai, P. Zhu, Z. Chang, Q. Wu, Y. Zhao, Y. Jia, P. Xu, H. Liu, and G. Shan. "Exon-Intron Circular Rnas Regulate Transcription in the Nucleus." Nat Struct Mol Biol 22, no. 3 (2015): 256-64"; DOI: 10.1038/nsmb.2959.

Point 9: Sentence in Line 151 needs rephrasing.

Response 9: We have corrected this sentence in our revised manuscript.

Point 10: Change MiR-148b-3p to miR-148b-3p in Table 2.

Response 10: We have changed MiR-148b-3p to miR-148b-3p in Table 2 in the revised manuscript.

Point 11: Reference 84 should be excluded from Table since the authors only include studies based in the last 3 years.

Response 11: We have removed reference 84 from Table in the revised manuscript.

Major comments:

Point 12: The English of this manuscript can be significantly enhanced with additional proofreading.

Response 12: We have already extensively revised our English language grammar and spelling through the language editing service offered by MDPI in the revised manuscript.

Point 13: The authors need to highlight whether the reported literature are clinical, in vivo, or in vitro studies. It will be useful to include them in the Tables shown.

Response 13: From the literatures we discussed in this review, we noticed that most studies on the miRNA sponge functions of circRNAs in BC and CRC primarily described the indirect relationship between the circRNA–miRNA axis and malignant behaviors from the vivo or vitro studies. However, very few or even no circRNAs were used as the clinical diagnosis or therapy biomarkers in BC and CRC. So we kept the original description in the revised manuscript.

Point 14: There are circRNAs that can undergo translation. This needs to be discussed and highlight any articles that are involved in these cancers.

Response 14: We have added a description that circRNAs undergo translation in the section of “5.2. Additional Roles of CircRNA in BC and CRC” in the revised manuscript and also summarized in figure 2.

Point 15: Also, it will be informative to include if the circRNAs are upregulated or downregulated in the reported research, if any, into the Tables.

Response 15: In the text, we describe circRNAs as "oncogene and tumor-suppressor gene", and the expression level of oncogene circRNA was upregulated, while the expression level of tumor-suppressor circRNA was downregulated during the initiation and development of BC and CRC. So, we added this summarize in our revised manuscript.

Point 16: Figure 1 needs more work. The cartoons are not easily understood, and legends are not self-sufficient. The authors need to identify the different cartoons shown. E.g splice sites, blue pacman, etc. illustration needs to be clearer with lesser colors used.

Response 16: We have simplified figure 1 and added the related signaling pathways involved in the miRNA sponge function of cicRNA in BC or CRC in the revised manuscript.

Point 17: The authors need to emphasis or suggest various methods on leveraging circRNAs in the treatment or prognosis of BC and CRC. For example, there has been a few research working on artificial circRNAs to sponge specific miRNAs or proteins. These suggestions can be, if possible, included in Figure 1, to show readers on the different pathways that can be exploited to treat or prognose the disease.

Response 17: We have added some signal pathways involved in miRNA sponge in Figure 1 in the revised manuscript. However, great deal of studies on the miRNA sponge functions of circRNAs in BC and CRC primarily described the indirect relationship between the circRNA–miRNA axis and malignant behaviors, or some signal pathway involved in this relationship between circRNA–miRNA axis and malignant phenotypes in BC and CRC. Very few or even no artificial circRNAs were used as the clinical diagnosis or therapy biomarkers in BC and CRC.

Point 18: There are some biomarkers that used circRNA for prognosis. This information is useful to be included.

Response 18: From the literatures we discussed in this review, we noticed that most studies on the miRNA sponge functions of circRNAs in BC and CRC primarily described the indirect relationship between the circRNA–miRNA axis and malignant behaviors theoretically. However, very few or even no circRNAs were used as the clinical diagnosis or therapy biomarkers in BC and CRC.

Round 2

Reviewer 3 Report

I am satisfied with the revised manuscript.